# Epidemic modelling of monitoring public behavior using surveys during pandemic-induced lockdowns

Andreas Koher[1], Frederik Jørgensen [2], Michael Bang Petersen [2] & Sune Lehmann [1,3✉]

## Abstract

**Background** Implementing a lockdown for disease mitigation is a balancing act: Non-pharmaceutical interventions can reduce disease transmission significantly, but interventions also have considerable societal costs. Therefore, decision-makers need near real-time information to calibrate the level of restrictions.

**Methods** We fielded daily surveys in Denmark during the second wave of the COVID-19 pandemic to monitor public response to the announced lockdown. A key question asked respondents to state their number of close contacts within the past 24 hours. Here, we establish a link between survey data, mobility data, and hospitalizations via epidemic modelling of a short time-interval around Denmark's December 2020 lockdown. Using Bayesian analysis, we then evaluate the usefulness of survey responses as a tool to monitor the effects of lockdown and then compare the predictive performance to that of mobility data.

**Results** We find that, unlike mobility, self-reported contacts decreased significantly in all regions before the nation-wide implementation of non-pharmaceutical interventions and improved predicting future hospitalizations compared to mobility data. A detailed analysis of contact types indicates that contact with friends and strangers outperforms contact with colleagues and family members (outside the household) on the same prediction task.

**Conclusions** Representative surveys thus qualify as a reliable, non-privacy invasive monitoring tool to track the implementation of non-pharmaceutical interventions and study potential transmission paths.

### Plain language summary

Mobile phone data obtained from companies such as Google and Apple have often been used to monitor public compliance with pandemic lockdowns and make predictions of future disease spread. Survey data obtained by asking people a series of questions can provide an alternative source of information. We undertook daily surveys of a representative subset of the Danish population immediately before, and during, a lockdown during the COVID19 pandemic. We compared the modeling results obtained from the surveys with data derived from the movement of mobile phones. The self-reported survey data was more predictive of future hospitalizations due to COVID than mobility data. Our data suggest that surveys can be used to monitor compliance during lockdowns.

[1] DTU Compute, Technical University of Denmark, Lyngby, Denmark. [2] Department of Political Science, Aarhus University, Aarhus, Denmark. [3] Center for Social Data Science, University of Copenhagen, Copenhagen, Denmark. ✉email: sljo@dtu.dk

Pandemic management is a balancing act. When an outbreak of infections flares up, governments and authorities need to impose restrictions and recommendations on society that are carefully calibrated to the situation. On the one hand, during the COVID-19 pandemic, such non-pharmaceutical interventions have considerable benefits by changing the dominant transmission route—close contacts between individuals—via the incentives and information they provide[1,2]. On the other hand, these interventions have considerable costs in the form of negative externalities relating to the economy and mental health[3–5].

This balancing act puts authorities and governments in need of information to continuously calibrate the level of restrictions. It is not a matter of simply sending out a single set of instructions regarding restrictions and recommendations. Rather, authorities need to continuously receive information about the effectiveness of those restrictions and recommendations and adjust accordingly. An obvious source of information is directly related to the epidemic and includes the number of infection cases, hospitalizations, and deaths. Yet cases of infection are difficult to monitor and, for example, changes in the public's motivation to participate in testing programs may create problems with respect to comparisons over time[6]. Furthermore, there is a significant lag between the onset of interventions and hospitalizations and death counts, which imply that it is difficult to calibrate interventions on the basis of such information. Consequently, researchers, authorities and governments worldwide have complemented epidemiological information with information on the direct target of the interventions: Behaviour[7,8].

In this manuscript, we assess the predictive performance of a particular source of information about behavior during lockdowns: Population-based surveys on social contacts, fielded daily to representative samples of the Danish population during the COVID-19 pandemic (see *Methods* for details on this dataset). This assessment aligns with recommendations about the use of surveys as epidemic monitoring tools on the basis of experiences during the SARS epidemic in Hong Kong[9] and recommendations from the World Health Organization during the COVID-19 pandemic[10]. From a public health policy perspective, this particular dataset is a unique test case as it was, in fact, reported to the Danish government for this purpose on a twice-weekly basis during the second wave of COVID-19 infections in December 2020.

Furthermore, these data are unique in another respect: They constitute an open and 'citizen science'[11] alternative to the most used source of information on pandemic behavior: Mobility data. As we detail below, mobility data as a source of information may be problematic from both a methodological and policy perspective. Mobility data provides a proxy for close contacts between people and has been heavily utilized by researchers and public health institutions[8,12–14]. Mobility data quantifies the population's movement patterns and is unobtrusively obtained in a number of ways, for example, via people's smart phones and provided to researchers and governments via private companies such as Google[15]. This reliance, however, can and has raised concerns. First, in many cases, it implies that pandemic management and research relies on the willingness of private companies to share information during a critical crisis. Second, citizens themselves may be concerned about real or perceived privacy issues related to the sharing of data with authorities[16,17]. Given the importance of public trust for successful pandemic management[18], such concerns—if widespread—can complicate pandemic control. Third, data from companies such as Google, Facebook and local phone companies may not be representative of the population of interest: The entire population of the country. Rather than being invited on the basis of traditional sampling methods, people opt-in to the services of different companies and, hence, the data from any single company is likely a biased sample. In this sense, we argue that the known unknowns of survey data (e.g. we know we do not observe anyone under the age of 18) is preferable to the unknown unknowns of large-scale passive surveillance data. Fourth, the movements of people in society as captured by mobility data is only a proxy of the quantity of interest: Actual close encounters between individuals that drive the pandemic.

For these reasons, it is key to assess alternative sources of information about public behavior such as nationally representative surveys of the adult population. In principle, surveys could alleviate the problems relating to the collection and validity of mobility data. Survey research is a centuries old low-cost methodology that can be utilized by public actors and that relies on well-established procedures for obtaining representative information on private behaviours in voluntary and anonymous ways[19].

At the same time, data from surveys come with their own methodological complications. As documented by decades of research, people may not accurately report on their own behaviour[20]. Survey answers during the pandemic may be biased by, for example, self-presentational concerns and inaccurate memory. While research on survey reports of behaviour during the pandemic suggests that self-presentational concerns may not affect survey estimates[21], memory biases may (although such biases are likely small for salient social behavior)[22]. Even with such biases, however, surveys may be fully capable to serve as an informative monitoring tool. The key quantity to monitor is change in aggregate behaviour over time. If reporting biases are randomly distributed within the population, aggregation will provide an unbiased estimate. Even if this is not the case, changes in the survey data will still accurately reflect changes in population behaviour as long as reporting biases are stable within the relevant time period.

On this basis, the purpose of the present manuscript is, first, to examine the degree to which survey data provide useful diagnostic information about the trajectory of behavior during a lockdown and, second, to compare its usefulness to information arising from mobility data. To this end, we focus on a narrow period around Denmark's lockdown during the second wave of the COVID-19 epidemic in the Fall of 2020, i.e., prior to vaccine roll-out when it was crucial for authorities to closely monitor public behavior. We illustrate the usefulness of survey data on a narrow window of time because the changing nature of factors such as seasonal effects, new variants, vaccines, changing masking efforts, etc., make it difficult to model COVID-19 transmission across long periods without making a large number of assumptions[6]. See also Sec. 3 for a discussion on the limitations of our survey data. In spite of the limited scope, we believe that the study remains relevant for policy makers because it allows to monitor public behaviour at a crucial moment, when policy makers should not be forced to rely on proximity or mobility data from private companies in the absence of timely incidence data.

Specifically, we ask whether (a) daily representative surveys regarding the number of close social contacts and (b) mobility data allow us to track changes in the observed number of hospitalizations in response to the lockdown. In addition, to further probe the usefulness of survey data, we provide a fine-grained analysis of how different types of social contacts relate to hospitalizations. Our results shed new light on the usefulness of survey data. Previous studies during the COVID-19 pandemic have documented high degrees of overlap between self-reported survey data on social behavior and mobility data, but have not assessed whether these data sources contain useful information for predicting transmission dynamics[23,24]. One study did compare the predictive power of mobility data to survey data on the

psychosocial antecedents of behavior[25] and found that mobility data was more predictive than the survey data of COVID-19 transmission dynamics. Here, we provide a more balanced test by comparing the predictive value of mobility data and survey data when directly focused on self-reported behavior rather than simply its psychosocial antecedents.

We find that, unlike mobility, self-reported contacts decreased significantly in all regions of Denmark before the nation-wide implementation of non-pharmaceutical interventions. This change in behaviour corresponds well to the inferred reproduction number suggesting that self-reported survey data can be used to monitor compliance during lockdowns and improve short-term predictions of future hospitalizations. Further analyses of contact type show that contacts to friends and strangers outperform contacts with colleagues and family members (outside the household) as predictors for future hospitalization.

## Methods

**Data**. We use survey data from the HOPE ('How Democracies Cope With COVID-19') research project (www.hope-project.dk). Specifically, the HOPE-project fielded daily nationally representative survey in Denmark starting from mid-May 2020. Kantar Gallup, a private company, conducted the data collection until March 2022. Each day a nationally representative sample (with a daily target of 500 complete interviews) reports on their protective behaviour and perceptions of the COVID-19 pandemic. Participants are Danish citizens aged 18 years or older. They are recruited using stratified random sampling—on age, sex and geographical location—based on the database of Danish civil registration number. The data collection fully complies with Aarhus University's Code of Conduct and with the ethical standards set by the Danish Code of Conduct for Research Integrity. The legal aspects of the data collection was approved by Aarhus University's Technology Transfer Office. As per section 14(2) of the Act on Research Ethics Review of Health Research Projects, "notification of questionnaire surveys … to the system of research ethics committee system is only required if the project involves human biological material." All participants provided informed consent. The mobility data comes from Apple[26], Google[27] and major Danish mobile phone network operators[28]. For further details on the data, see Supplementary Note 1 and Supplementary Note 2.

**Model description**. We observe regional COVID-19 related hospitalizations, which derive from an initial number of infected and the time-varying reproduction number. We parametrize the latter using behavioural survey data and mobility time series. Our approach is a variant of the semi-mechanistic hierarchical Bayesian model of Flaxman et al.[29] and Unwin et al.[30], with the key difference that we use daily COVID-19 related hospitalizations. In Denmark, hospitalizations are a reliable proxy for pandemic activity available. Unlike death counts, hospitalizations are recorded with a significantly smaller delay and give a better signal-to-noise ratio for regions with little epidemic activity. The number of positive PCR-cases, on the other hand, suffers from confounding through varying test intensity during the Christmas holidays and more importantly, we can rely on a well-studied infection-to-hospitalization delay distribution, which is less sensitive to country-specific testing protocols.

The code is written in the Julia programming language[31] using the Turing.jl package[32] for Bayesian inference. The source code is fully accessible on GitHub[33] and we summarize sampling details in Supplementary Note 3. In the following, we provide the mathematical details of the epidemiological model.

*Observation model*. As observations, we take the daily number of hospitalizations $H_{t,r}$ at time $t$ in region $r$ and assume these are drawn from a Negative Binomial distribution with mean $h_{t,r}$ and over-dispersion factor $\phi$:

$$H_{t,r} \sim \text{NegBinom}\left(h_{t,r}, h_{t,r} + \frac{h_{t,r}^2}{\phi}\right) \quad (1)$$

$$\phi \sim \text{Gamma}(\text{mean} = 50, \text{std} = 20) \quad (2)$$

From the expected number of hospitalizations $h_{t,r}$, we derive the latent, i.e., unobserved number of new infections $i_{t,r}$. Two factors link infections to hospitalizations: (a) The conditional probability $\alpha$ of hospitalization following an infection and (b) the corresponding delay distribution $\pi$:

$$h_{t,r} = \alpha \sum_{\tau=0}^{t-1} i_{\tau,r} \pi_{t-\tau} \quad (3)$$

$$\alpha \sim \text{Normal}^+(0.028, 0.002) \quad (4)$$

We estimate the infection hospitalization rate $\alpha$ in Eq. (4) from a sero-prevalence study[34]. The results are, however, not sensitive to this value as we don't account for the depletion of susceptible. The delay $\pi$ is a sum of two independent random variables, i.e. the incubation period and the time from onset of infection to hospitalization[35]. We take the corresponding distributions from previous studies and parametrize the incubation period by a Gamma distribution with a mean of 5.1 days and a coefficient of variation of 0.86[36] and the infection to hospitalization delay by a Weibull distribution with a mean of 5.506 days and a shape parameter 0.845[35], which corresponds to a standard deviation of 8.4 days:

$$\pi \sim \text{Gamma}(\text{mean} = 5.1, \text{CV} = 0.86)$$
$$+ \text{Weibull}(\text{shape} = 0.845, \text{scale} = 5.506) \quad (5)$$

We then discretize the continuous distribution $\pi$ by $\pi_i = \int_{i-0.5}^{i+0.5} g(\tau)d\tau$ for $i = 2, 3, \ldots$ and $\pi_1 = \int_0^{1.5} g(\tau)d\tau$ for application in Eq. (3).

*Infection model*. The (unobserved) number of new infections, $i_{t,r}$, evolves according to a discrete renewal process. This approach has been widely used in epidemic modelling[29,37–39], is related to the classical *susceptible-infected* model[40] and has a theoretical foundation in age-dependent branching processes[37,41]. New infections in region $r$ at time $t$ are a product of the time-varying reproduction number $R_{t,r}$ and the number of individuals that are infectious at time $t$. The latter is a convolution of past infections and the generation interval $g_\tau$:

$$i_{t,r} = R_{t,r} \sum_{\tau=0}^{t-1} i_{\tau,r} g_{t-\tau} \quad (6)$$

The generation interval $g$ translates past infections to the present number of infectious individuals and following previous studies, we assume a Gamma distribution density $g(\tau)$ with mean 5.06 and SD 2.11[42]:

$$g \sim \text{Gamma}(\text{mean} = 5.06, \text{SD} = 2.11) \quad (7)$$

Again, we discretize the continuous distribution by $g_i = \int_{i-0.5}^{i+0.5} g(\tau)d\tau$ for $i = 2, 3, \ldots$ and $g_1 = \int_0^{1.5} g(\tau)d\tau$ to be used in Eq. (6). The convolution in Eq. (6) requires a history of infectious individuals for initialization, which we estimate prior to the analysis as described below.

*Transmission model*. At the heart of the analysis is the instantaneous reproduction number $R_{t,r}$ for region $r$ at time $t$. It

determines the number of secondary transmissions from the current number of infectious individuals. We implement a *parametric* and a *non-parametric* variant of the model akin to[43].

The *non-parametric model* implements a latent random-walk, i.e., a AR(1) process that allows to track daily changes of the reproduction number:

$$R_{t,r} = R_{0,r} \exp(\rho_{t,r}) \tag{8}$$

$$\rho_{t,r} \sim \text{Normal}(\rho_{t-1,r}, \sigma) \tag{9}$$

$$\sigma \sim \text{Normal}^+(0.3, .02) \tag{10}$$

Here, the latent variable $\rho_{t,r}$ performs a random walk with a typical step size of $\sigma$. Hence, the number of inferable parameters $\rho_{t,r}$ equals the number of observation days for each region $r$. The step size $\sigma$ determines the smoothness of the resulting reproduction number and we choose the same prior distribution as in ref. [30]. The non-parametric model allows us to infer the "ground truth" that we use for visual comparison.

The *parametric model*, on the other hand, takes a data stream $X_{t,r}$ for every region $r$ as a parametrization of the reproduction number:

$$R_{t,r} = R_{0,r} \exp(e_r X_{t,r}) \tag{11}$$

$$e_r \sim \text{Normal}(e, s) \tag{12}$$

$$e \sim \text{SkewedLaplace}(\mu = 0, \sigma = 0.7, \alpha = 0.2) \tag{13}$$

$$s \sim \text{Gamma}(\text{mean} = 0.07, \text{SD} = 0.05) \tag{14}$$

The predictors are normalized such that $X_{t,r}$ gives the change in behaviour at time $t$ relative to the first day, i.e. $t_0 = 2020\text{-}12\text{-}01$, in region $r$. Thus, the effect size $e_r$ in Eq. (11) translate a relative change in the predictor $X_{t,r}$ to a change in the regional reproduction number $R_{t,r}$. We pool information in order to reduce regional biases and to give a robust country-level effect estimate $e$ akin to multi-level models[44].

With more contacts or a higher mobility level, we expect an increased disease transmissibility and therefore, we choose a skewed Laplace distribution as a prior for the pooled effect parameter $\mu_e$[45]. Furthermore, we choose a shrinking prior on the dispersion parameter $s$ to limit regional differences and thus reduce potential overfitting given the limited data. Note, however, that substantial effect differences are still inferrable if the data provides sufficient evidence.

*Initialization of the non-parametric model.* Observations start on 01-August-2020, i.e., well before the second wave of Covid infections (see Fig. 2). In order to initialize the discrete renewal process, we can therefore reasonably assume that the number of latent infections prior to 01-August-2020 are constant, i.e., $i_{t,r} \equiv i_{0,r}$ for $t \leq 0$. We infer $i_{0,r}$ from the number of PCR-positive cases $I_{0,r}$ on 01-August-2020 and roughly assume an underestimation factor of three:

$$i_{0,r} \sim \text{Exponential}(3 I_{0,r}) \tag{15}$$

The exponential prior implies a broad uncertainty and thus sufficient flexibility of the inference model. Note that we choose PCR-positive cases to initialize the number of infected because hospitalizations were very low and noisy at the start of the second wave, making incidence data in this case a stronger choice for initializing the model. Moreover, we choose the initial reproduction number to be around one, which reflects our prior believe that the epidemic was under control well before the second wave

of infections:

$$R_{0,r} \sim \text{Normal}^+(1.0, 0.1) \tag{16}$$

*Initialization of the parametric model.* Observations start on 01-December-2020, i.e., about 1 week prior to the lockdown's announcement and well withing the second wave of Covid-19 infections. Here, the assumption of constant $i_{t,r} \equiv i_{0,r}$ for $t \leq 0$ as well as $R_{0,r} \approx 1$ are not suitable. Instead, we take posterior samples from the non-parametric model, marked with an asterisk, for initialization: In particular, we take the mean over the posterior samples of the latent infections $\langle i \rangle_{t,r}^*$ and scale the timeseries with a factor $\nu$ that corresponds roughly to the posterior uncertainty of $i_{t,r}^*$. Hence, we obtain the initial number of latent infections according to:

$$i_{t,r} = \nu \cdot \langle i \rangle_{t,r}^* \quad \text{for all } t \leq 0 \tag{17}$$

$$\nu \sim \text{Normal}^*(1, 0.1) \tag{18}$$

Similarly, we initialize the effective reproduction number $R_{0,r}$ by fitting a Normal distribution to the posterior samples $R_{0,r}^*$ from the non-parametric model at the initial observation, i.e. 01-December-2020:

$$R_{0,r} \sim \text{Normal}^+(\mu_R, \sigma_R) \tag{19}$$

$$\mu_R = \text{mean}(R_{0,r}^*) \tag{20}$$

$$\sigma_R = \text{std}(R_{0,r}^*) \tag{21}$$

*Parametric model with multiple predictors.* For the analysis in Supplementary Fig. 1 and Supplementary Table 1, we implement a parametric model with multiple predictors $c$. To this end, we modify Eq. (11) to Eq. (14) according to:

$$R_{t,r} = R_{0,r} \exp\left(\sum_c e_r^c X_{t,r}^c\right) \tag{22}$$

$$e_r^c \sim \text{Normal}(e^c, s) \tag{23}$$

$$e^c \sim \text{SkewedLaplace}(\mu = 0, \sigma = 0.7, \alpha = 0.2) \tag{24}$$

$$s \sim \text{Gamma}(\text{mean} = 0.07, \text{SD} = 0.05) \tag{25}$$

The reproduction number in region $r$ at time $t$ is a linear combination multiple data streams $X_{t,r}^c$ with an exponential link-function to ensure positivity. Each predictor is normalized such that $X_{t,r}^c$ gives the change in behaviour or mobility at time $t$ relative to the first day, i.e. 2020-12-01, in region $r$. Thus, the effect sizes $e_r^c$ translate a relative change in the predictor $c$ to a change in the reproduction number $R_{t,r}$. We pool effect sizes $e_r^c$ to reduce regional biases and obtain a national-level effect size $e^c$ for each predictor $c$.

**Reporting summary**. Further information on research design is available in the Nature Portfolio Reporting Summary linked to this article.

## Results

We establish the link between survey data, mobility data, and hospitalizations via epidemic modeling, which uses the behavioural survey and mobility data as an input to capture underlying infectious activity[30,43]. Specifically we extend the semi-mechanistic Bayesian model from Flaxman et al.[29,30] to jointly model the epidemic spreading within the five regions of Denmark. Where possible, we use partial pooling of parameters to share information across regions and thus reduce region specific biases. We parametrize the regional reproduction number $R_t$ with

a single predictor $X_t$ from our survey or the mobility data, respectively, for each realization of a model:

$$\log(R_t) = \log(R_0) + eX_t \qquad (26)$$

The regional reproduction number at time $t$ derives from the initial value $R_0$ and the scaled predictor $eX_t$ with a logarithmic link-function (see *Methods* for full details on the model).

We compare the predictive performance of each data stream using leave-one-out cross-validation (LOO). LOO works by fitting the model to the observed hospitalizations excluding a single observation and comparing the prediction of the unseen observation against the observed real-world data. Repeating this process over all observations, allows one to estimate the model performance on out-of-sample data with a theoretically principled method that accounts for uncertainties[44]. In practice, this would result in an immense computational effort and therefore, we use an efficient estimation of LOO based on pareto-smoothed importance sampling[46]. In order to compare the predictive performance of, say self-reported survey against mobility, we calculate the LOO score for each model parametrization and consider the difference significant if it exceeds the 95% CI.

Because we are interested in the use of behavioural data as a guide for decision-making, our inference focuses on the key period of the second wave from 1-December-2020, i.e., about 1 week before Denmark's lockdown announcement, to 20-February-2021 when vaccinations accelerated across the country. The period captures a sharp increase and eventual decline in hospitalizations during the second wave of Denmark's Covid-19 pandemic (see Supplementary

Fig. 2). We stress that this narrow focus makes ours a proof-of-concept study. To fully understand the efficacy of survey-data, it will be important to extend models and analyses to longer periods of time—thus making it necessary to involve factors such as vaccination, new variants of concern, the opening of schools, etc.

**Defining risk-taking behaviour.** As a monitoring tool, we first consider self-reported survey data on the daily number of contacts, defined as close encounters with <2 meters distance for at least 15 minutes[47]. The reported numbers are highly skewed, with 15.7% of all counts concentrated on zero with some reporting over 1000 contacts (see Supplementary Fig. 3). As a result, taking the mean over daily reported numbers is highly sensitive to outliers, while reporting quantile-based measures obscure most of the variation.

Instead, we define the following robust measure of risk-taking behaviour: We label a participant in the survey as risk-taking if they report contacts above a fixed threshold and propose the daily *fraction of risk-taking* individuals as a predictor to the effective reproduction number. The intuition is that infections tend to be linked to large clusters via super-spreading events[48]. Therefore, we base our analysis on the fraction of the population that reports an above-average number of contacts.

That choice begs the question 'What is a reasonable threshold that defines risk-taking behaviour?' We choose a reference period prior to the lockdown's announcement, take the distribution of contacts over the time window and define a range of thresholds in terms of percentiles (see Supplementary Fig. 3 for details). For a visual comparison, Fig. 1, second row illustrates the dynamics of

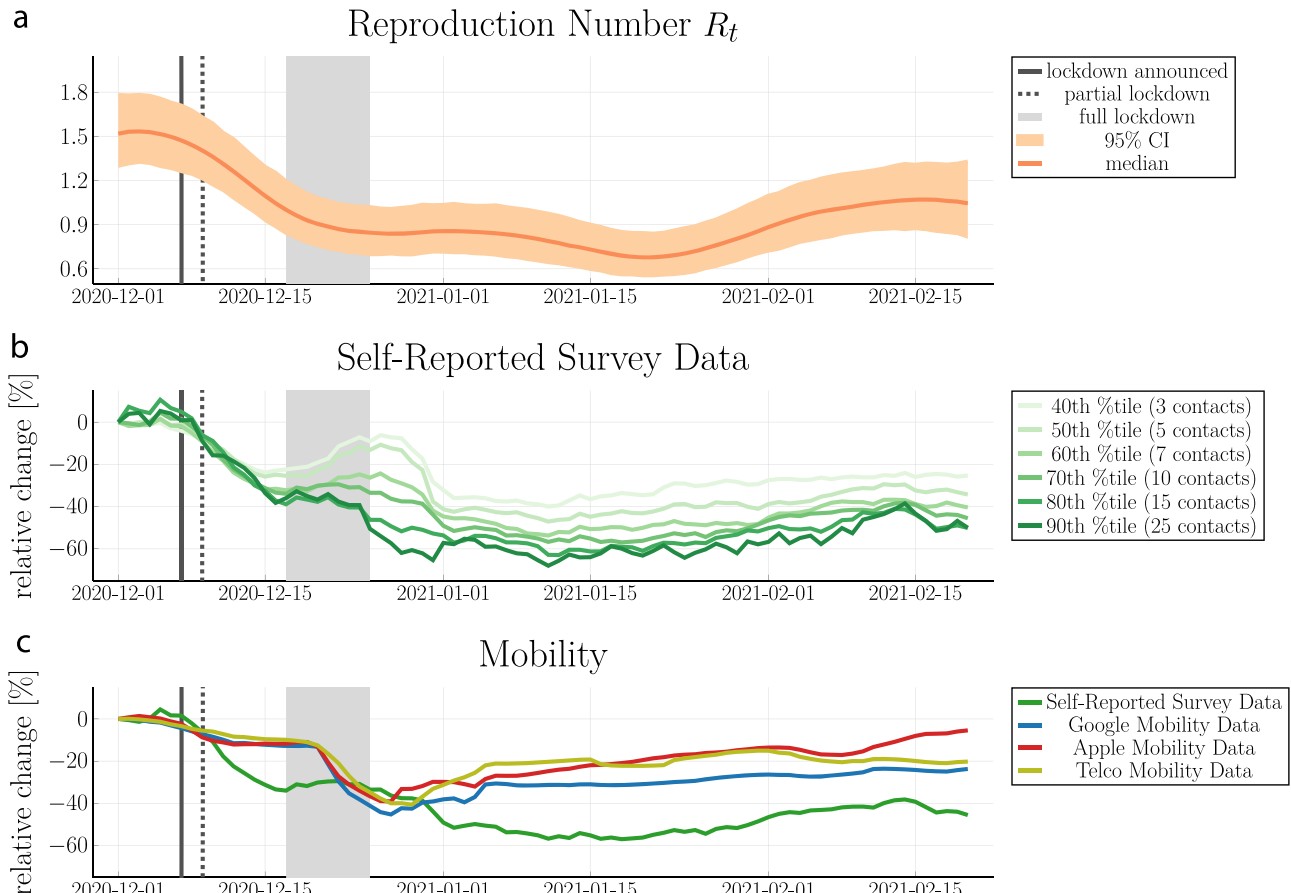

**Fig. 1 National-level comparison between the inferred reproduction number and multiple predictors. a** inferred reproduction number from national hospitalizations. **b** Comparison between thresholds that define risk-taking behaviour: The percentile gives a number of contacts $n$ that defines risk-taking behaviour. The time-series present the daily fraction of individuals $P(\#\text{total contacts} \geq n)$ that report at least $n$ contacts. **c** Comparison between risk-taking behaviour with a threshold at the 70th percentile (*self-reported survey data*), Google mobility, Apple mobility, and telecommunication data (*Telco*).

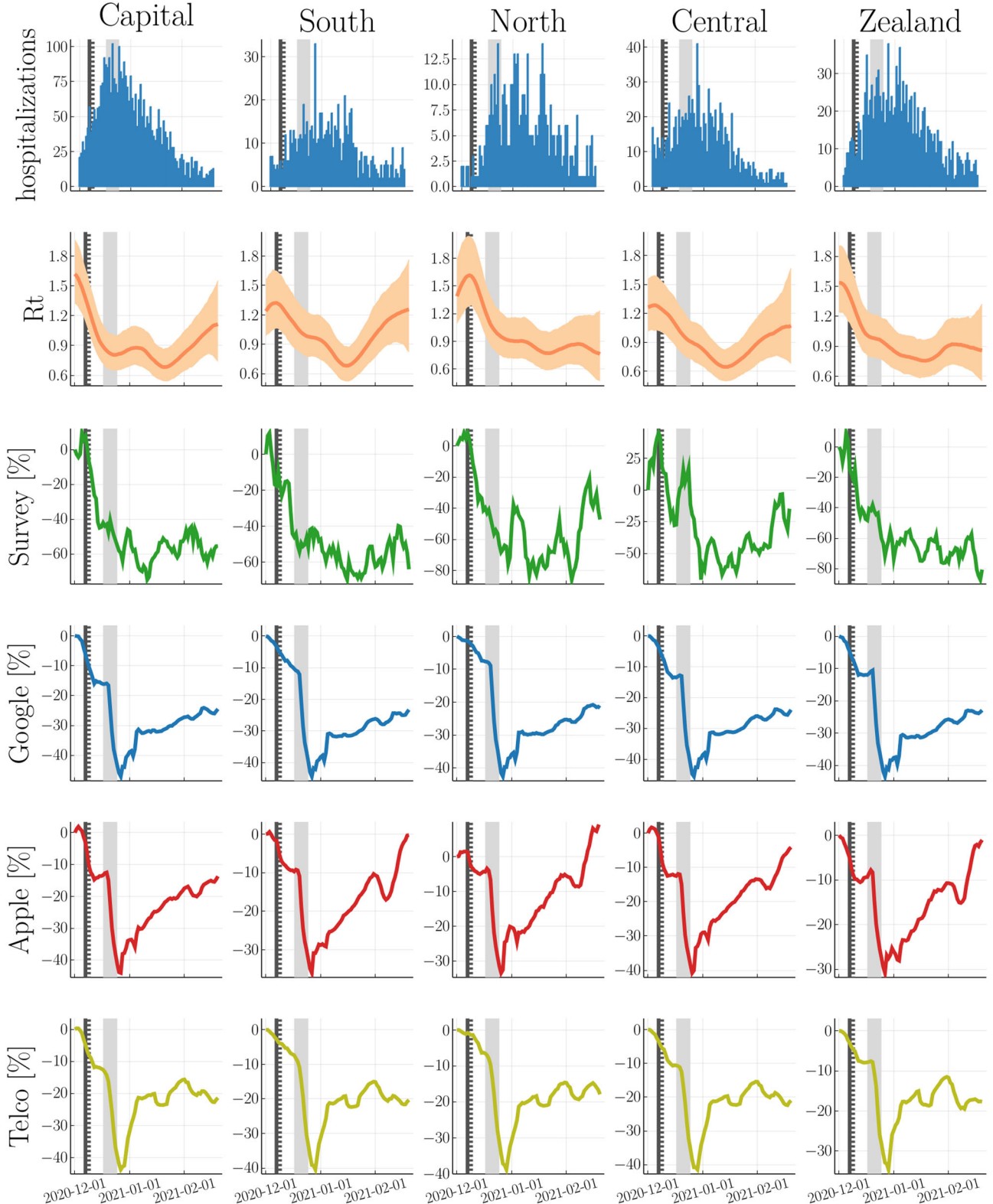

**Fig. 2 Regional-level comparison between hospitalizations, reproduction number and predictors.** First row: Hospitalizations. 2nd row: inferred reproduction number from regional hospitalizations with mean and 95% CI. 3rd-6th row: survey data (70th percentile threshold), Google mobility, Apple mobility, and telecommunication data (*Telco*). We mark the lockdown's first announcement, it's partial and national implementation with a solid vertical line, a dashed vertical line and shaded vertical area, respectively.

risk-taking behaviour, referred to as *self-reported survey data*. The thresholds range from the 40th to the 90th percentile and translate into a critical number of contacts ranging from 3 and 25, respectively. For thresholds above the 60th percentile, risk-taking

behaviour shows the strongest response to the announced lockdown and increases little during the Christmas period. Qualitatively, this behaviour matches the time-varying reproduction number $R_t$ (see Fig. 1, first row) that we inferred from

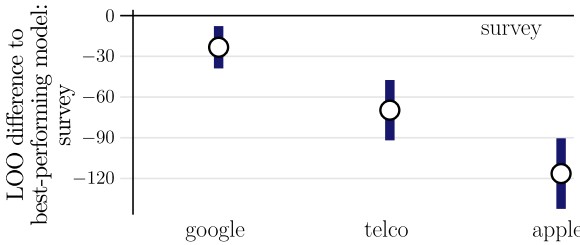

**Fig. 3 Self-reported survey data (*survey*) demonstrates highest predictive performance compared to Google mobility, Apple mobility and telecommunication data (telco).** We calculate the difference in LOO score w.r.t the best performing model and mark the mean difference and the 95% CI with a circle and a blue bar, respectively. We consider the difference significant if the mean exceeds the 95% CI. See Supplementary Table 4 for details.

national hospitalizations using a latent random-walk model (details in Sec. 3).

In the following, we use the 70th percentile as a threshold, which corresponds to 10 close contacts and more within the past 24h. However, our results are not sensitive to this value as all models within a threshold between the 60th and 90th percentile perform similarly well (see Supplementary Fig. 4 and Supplementary Table 2).

**Self-reported survey data versus mobility data.** By considering self-reported survey data, we capture the sharp decline in the reproduction number after the lockdown's announcement, i.e., about 2 weeks before its nationwide implementation. This early signal is not as pronounced in the combined mobility time series from Google and Apple that have been proposed in ref. [43], nor in the telecommunication data from Danish mobile network operators (see Fig. 1 and Fig. 2 for a visual comparison on the national and regional level, respectively). In addition, we also observe a sharp increase in mobility shortly after the lockdown's implementation, which does not correspond to the inferred reproduction number and thus does not translate into increased hospitalizations. This decoupling between mobility and disease dynamics has been previously observed for other countries[43,49]. A quantitative model comparison with LOO cross-validation confirms that self-reported survey data gives the best out-of-sample predictions for hospitalizations (see Fig. 3).

We find a more nuanced result when comparing self-reported contacts to the individual data streams provided by Google (see Supplementary Fig. 5). In particular, the category "Retail & Recreation" performs only marginally worse (see Supplementary Table 3) suggesting that disease relevant contacts are highly context dependent—a result that we will examine in the following section.

**Understanding the role of contact-types.** In our survey, we assessed the daily number of contacts separately for (a) family members outside the household, (b) friends and acquaintances, (c) colleagues and (d) strangers, i.e. all other contacts. Therefore, we can evaluate the impact of social context-depending risk-taking behaviour on $R_t$ and observed hospitalizations, respectively (Fig. 4). As above, we choose the 70th percentile as a threshold for risk-taking behaviour for each contact type, and as above our findings are robust to the specific choice of threshold.

The visual comparison in Fig. 5 shows that risk-taking behaviour towards friends, strangers and colleagues declines significantly weeks before the lockdown's national implementation—unlike risk-taking behaviour towards family members. The latter spikes around Christmas, which appears to have little effect

on the reproduction number, perhaps due to precautionary measures taken prior to visiting family (e.g., testing).

Cross-validation shows that risk-taking behaviour towards friends and strangers is significantly more predictive than family members and colleagues (see Fig. 5). Importantly, however, this does not imply that contacts with colleagues and family members play a minor role in disease spreading. A joint model that includes all contact types as predictors reveals a strong correlation between risk-taking behaviour towards colleagues and family members (see Supplementary Fig. 6) and Supplementary Fig. 1 and a further cross-validation analysis shows that the combination of both predictors performs similarly well to contacts with strangers and friends (see Supplementary Table 1).

## Discussion

During a lockdown, decision-makers need high-fidelity, real-time information about social behavior in order to carefully calibrate restrictions to both the epidemic wave and levels of public compliance. Interventions that are too lenient will not sufficiently reduce the wave, while too severe interventions (e.g., curfews) may have significant negative externalities on, for example, public trust and mental health[4,5].

To this end, researchers and authorities worldwide have relied on mobility data, which have been cheaply available as they were already unobtrusively collected by, for example, private tech companies. At the same time, such reliance entails a dependency on data collected by company actors and data which may raise privacy issues.

In the present analysis, we have provided evidence suggesting the usefulness of daily surveys of nationally representative samples as an alternative source of information during a lockdown. While the use of surveys has been recommended during the COVID-19 pandemic by WHO[10] and on the basis of the SARS epidemic in Hong Kong[9], the present analysis provides one of the first attempts to quantify the predictive validity of surveys of self-reported behavior during a lockdown. In contrast, prior research has focused on the behavioral antecedents of behavior such as self-reported fear of COVID-19[25]. While understanding the impact of such antecedents is a theoretically important endeavour, more direct measures of behavior may be preferable for a monitoring purpose (see also Supplementary Fig. 7 and Supplementary Table 6 for a comparison with indirect measures from our survey).

Our analyses provides a proof-of-concept that self-reported measures of behavior can be superior to mobility. Given the widespread use of mobility data it is relevant to ask why survey data fared better. Unlike the telco data and the combined time-series from Google and Apple, respectively, the survey data was able to capture behavioural changes weeks before the lockdown's nation-wide implementation. Parts of the effect can be explained by preceding partial lockdowns (see Supplementary Table 7 for a timeline of Covid19 related restrictions). However, we see similar decreases of activity also in regions that were not targeted with the partial lockdown and in addition, we observe an early increase in risk-awareness (see Supplementary Fig. 8). This observation hints at an additional indirect, i.e., psychological effect: Individuals adjust their behaviour in response to an increased perceived threat due to rising case numbers or intensified political discussions that culminated in the announced national lockdown on 07-December-2020. This finding suggests that part of the problem of mobility data may be that it is too coarse and, hence, does not capture the micro-adjustments in social behavior that people make when they are concerned with infection risk such as standing further away from others in public queues, not mingling with co-workers at the workplace and so forth.

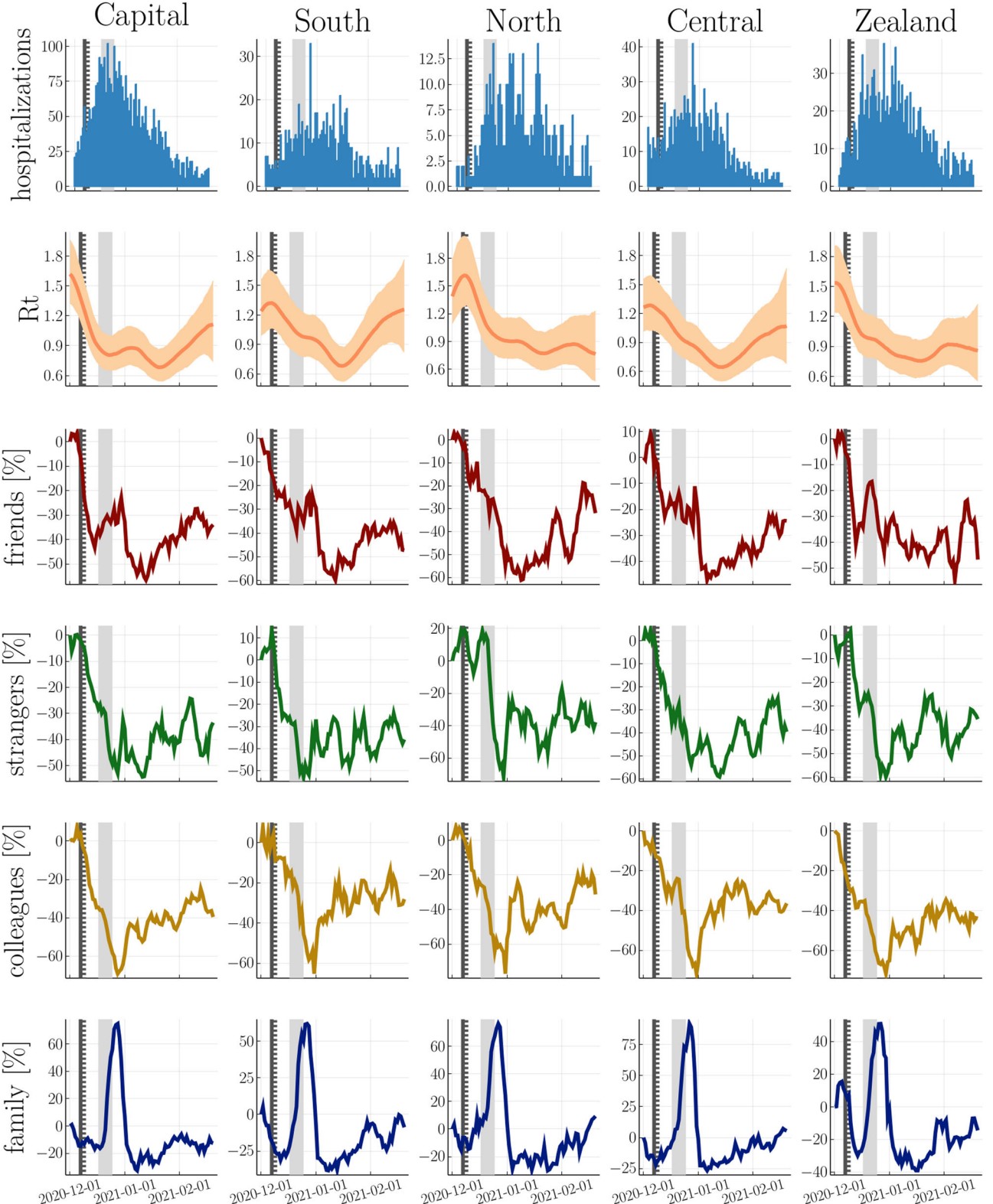

**Fig. 4 Regional-level comparison between hospitalizations, reproduction number and risk-taking behaviour in different social contexts.** 1st row: regions of Denmark. 2nd row: inferred reproduction number from regional hospitalizations with mean and 95% CI. 3rd-6th row: Regional predictors including risk-taking behaviour towards friends, strangers, colleagues, and family members outside the household, respectively, with a threshold at the 70th percentile. The solid vertical line, dashed vertical line and shaded area mark the lockdown's first announcement, it's partial implementation and national implementation, respectively.

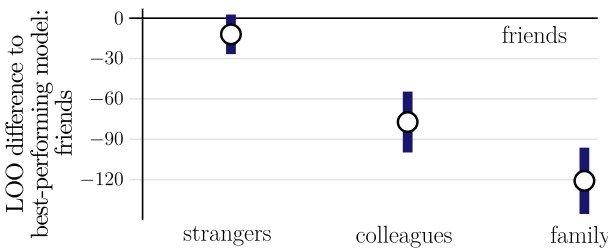

**Fig. 5 Risk-taking behaviour towards friends and strangers are the single best predictors for the observed hospitalizations.** We present the difference in LOO cross-validation w.r.t the best performing model and plot the mean and 95% CI as cicles and vertical bars. See Supplementary Table 5 for details. In Supplementary Table 1, we show that the combination of risk-taking behaviour towards colleagues and family members performs similarly well.

Moreover, mobility increases shortly after the lockdown's implementation with little effect on hospitalizations. This decoupling between mobility and reproduction number has been previously observed in other countries[43,49]. Unlike mobility, self-reported contacts provide a more direct measure of behaviour and thus improves predictability months after the lockdown's implementation.

At the same time, it is relevant to note that a more detailed analysis of the individual Google data streams revealed the importance of context-depending contacts: Our analysis finds that "Retail & Recreation" performs only marginally worse than self-reported contacts (see Supplementary Table 3) and Supplementary Fig. 5 and can be best explained by risk-taking behaviour towards strangers (see Supplementary Fig. 9).

Finally, we find that risk-taking behaviour towards strangers and friends provide the best predictors for hospitalizations, although, a joint model that includes contacts to colleagues and family members performs similarly well. This behaviour could be explained by their complementary dynamics during the Christmas period: Holidays implied less contacts to colleagues and larger gatherings with family members.

Our sensitivity analysis in Supplementary Figs. 10, 11, 12, 13, 14, and 15 confirms that all results are robust to minor changes in the observation window, the infection-to-hospitalization distribution, and the threshold that defines risk-taking behaviour.

Our inability to predict the rise of COVID-19 related hospitalizations prior to the lockdown's announcement suggests that there are multiple possibilities of improving the measures used for monitoring public behavior during an epidemic. When knowledge has been gathered about the main pathways of transmission, researchers and authorities can more directly ask questions about social interactions in situations that enhances or inhibits transmission risk. During the COVID-19 pandemic, for example, it would be relevant to know whether the contact occurred inside or outside, especially as temperatures drop and individuals adjust their behaviour. Moreover, we know now about the importance of transmission in children and young adults below 18, which could not be included in the study. We believe that the lack of contextual information and representativeness limits the usefulness of our data set to predict the onset of the second wave of COVID-19 infections. (see Supplementary Fig. 2).

An important final lesson is that if one is able to sample representatively, the surveys themselves do not need to be especially large—only around 500 responses per day in this case. However, this also raises an important limitation of our study. The samples we are able to collect in Denmark are arguably too good to transfer directly to other contexts. Our ability to sample directly from highly curated central database, with a 25%

response rate despite no compensation offered to respondents is not necessarily replicable in many other countries, especially outside of Europe[50]. We stress that the implications of reduced sample quality need to be explored if extending our results to other contexts.

In summary, the present analysis has provided proof-of-concept regarding the usefulness of survey data as public policy tool for monitoring compliance with the announcement and implementation of lockdowns. Even though, the analyses we present are narrowly focused on a single lockdown, they provide evidence in support for the WHO's recommendation to integrate social science methods such as surveys into pandemic surveillance and management.

## Data availability

All data necessary for the replication of our results is collated in https://github.com/andreaskoher/Covid19Survey[33]. This includes mobility data from Google (https://www.google.com/covid19/mobility/), Apple (https://covid19.apple.com/mobility), and Danish telco providers (https://covid19.compute.dtu.dk/data-description/telco_data/), as well as Covid-19 related hospitalizations in the five regions of Denmark (https://covid19.ssi.dk/). For convenience, we provide the regional hospitalization data together with the predictors used in the main text in Supplementary Data (see also https://doi.org/10.5281/zenodo.781879[51]).

## Code availability

All code necessary for the replication of our results is collated in https://github.com/andreaskoher/Covid19Survey[33].

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

## Acknowledgements
All authors are thankful to the Carlsberg Foundation who funded the study (Grant CF20-0044, HOPE: How Democracies Cope with Covid-19).

## Author contributions
A.K., F.J., M.B.P., and S.L. conceived the study and wrote the text. A.K. carried out modeling and analyses. F.J. and M.B.P. collected the survey data.

## Competing interests
The authors declare no competing interests.
