## [Peer Review File · Communications Medicine]

This manuscript has been previously reviewed at another Nature Portfolio journal. This document only contains reviewer comments and rebuttal letters for versions considered at Communications Medicine

REVIEWERS' COMMENTS:

Reviewer #1 (Remarks to the Author):

I appreciate methodological improvements and improvements in the presentation of the results. However, my main criticism of the analysis remains unchanged. My main comment was that authors needed to consider a longer time period to validate the statistical association they had identified between the reproduction number and their survey dataset. Instead, in their response, authors explain that they have simply narrowed the scope of their paper to the context of “lockdowns” only. I do not think that is a convincing argument. If there is indeed a functional relationship between R_t and contacts as investigated in the paper, there is no reason to believe that the relationship would only apply to lockdowns. I would argue that if it was only to apply to lockdowns, then the framework would be of limited value: lockdowns have been very well studied and we now have a good idea of the values of R we can expect during lockdowns, even without survey. The current analysis also does not address the fact that this is a small dataset describing dynamics during a short time period and so, it is hard to make definitive conclusions from it. I think a lot of readers would conclude from reading the paper that contact surveys are a good predictor of R_t , but I don't think the evidence presented is sufficient to say that.

In conclusion, I stand by my previous comment that the time period being studied is too short, even if authors explain that they only aim to study the impact of lockdowns.

[ED: we are over ruling this concern about the time period of study, following discussion with other experts].

Reviewer #2 (Remarks to the Author):

I'm late returning this review, and I was brought in for the most recent round of revisions, so I'll be brief: this is an excellent paper that will make an important contribution for both scholars and policymakers.

The overarching point that comes through in this paper is that despite surveys' well-known sources of biases and measurement error, these biases and errors are often no worse than those inherent to “ground truth” measures such as those derived from privately-collected cell phone data. While this doesn't necessarily need to be mentioned in the paper, I would add that the biases in survey data are closer to “known-unknowns” (we know we don't observe anyone under the age of 18 here, e.g.) than the “unknown-unknowns” of mobility data. Who opts out of mobility data collection, doesn't have a cell phone/app on their phone that would collect their mobility data, or leaves their cell phone at home when they go to the store, is a mystery — we just know the black box doesn't have everyone in it. It is therefore relatively intuitive that a well-executed survey can outperform these or other forms of administrative data when the task is to track over-time changes in behavior across the entire population and predict outcomes that are theoretically downstream of those behaviors.

I think it is also worth pointing out that, if you do the sampling well, the surveys themselves don't need to be especially large — only ~500 responses per day in this case. Here, I think this paper speaks nicely to Bradley, et al (2021), who are coming at this from the other direction

to push back on the use of (big, internet-based) surveys in this area. The issue is not the survey as a method itself, but the quality of the sample.

However, here is the one place where I would suggest the authors acknowledge one additional limitation: the samples they were able to collect in Denmark are arguably too good. The ability to sample directly from a government database, with a 25% response rate despite no compensation offered to respondents, is fantastic. It is also likely not replicable in many other countries, especially outside of Europe. To be clear, this is not to take away from what the authors have accomplished here, nor is it a reason to delay publication. It just isn't obvious to me that, practically speaking, anyone could get such high-quality survey data in the United States or other less developed countries. However, the question of whether similar results can be achieved with lower-quality survey data can be left for future work.

Reference

Bradley, Valerie, et al. 2021 "Unrepresentative big surveys significantly overestimated US vaccine uptake." *Nature* 600, 695-700.